


# Emissions of Organic Compounds from Western US Wildfires and Their Near Fire Transformations

Yutong Liang[1#], Christos Stamatis[2a#], Edward C. Fortner[3], Rebecca A. Wernis[4], Paul Van Rooy[2], Francesca Majluf[3], Tara I. Yacovitch[3], Conner Daube[3], Scott C. Herndon[3], Nathan M. Kreisberg,[5] Kelley
C. Barsanti[2], and Allen H. Goldstein[1,4]

[1]Department of Environmental Science, Policy, and Management, University of California, Berkeley, Berkeley, CA, 94720, USA

[2]Department of Chemical and Environmental Engineering and College of Engineering – Center for Environmental Research
and Technology (CE-CERT), University of California, Riverside, Riverside, CA, 92507, USA

[3]Aerodyne Research, Inc., 45 Manning Road, Billerica, MA, 01821, USA

[4]Department of Civil and Environmental Engineering, University of California, Berkeley, California 94720, USA

[5]Aerosol Dynamics, Inc., Berkeley, CA 94710, USA

[a]Now at Charles E. Via Jr. Department of Civil and Environmental Engineering, Virginia Tech, Blacksburg, VA, 24061, USA

#Y.L. and C.S. contributed equally to this work.

*Correspondence to*: Yutong Liang (yutong.liang@berkeley.edu)

**Abstract.** The size and frequency of wildfires in the western United States have been increasing and this trend is projected to
continue, with increasing adverse consequences for human health. Gas- and particle-phase organic compounds are the main
component of wildfire emissions. Some of the directly emitted compounds are hazardous air pollutants, while others can react
with oxidants to form secondary air pollutants such as ozone and secondary organic aerosol (SOA). Further, compounds
emitted in the particle phase can volatize during smoke transport and can then serve as precursors for SOA. The extent of
pollutant formation from wildfire emissions is dependent in part on the speciation of organic compounds. The most detailed
speciation of organic compounds has been achieved in laboratory studies, though recent field campaigns are leading to an
increase in such measurements in the field. In this study, we identified and quantified hundreds of gas- and particle-phase
organic compounds emitted from conifer-dominated wildfires in the western US, using two two-dimensional gas
chromatography coupled with time-of-flight mass spectrometry (GC×GC ToFMS) instruments. Observed emission factors
(EFs) and emission ratios are reported for four wildfires. As has been demonstrated previously, modified combustion efficiency
(MCE) was a good predictor of particle phase EFs, except for elemental carbon. Higher emissions of diterpenoids, resin acids
and monoterpenes were observed in the field relative to laboratory studies; likely due to distillation from unburned heated
vegetation, which may be underrepresented in laboratory studies. These diterpenoids and resin acids accounted for up to 45%
of total quantified organic aerosol, higher than the contribution from sugar and sugar derivatives. The low volatility of resin



acids makes them ideal markers for conifer fire smoke. The speciated measurements also show that evaporation of semi-volatile organic compounds took place in smoke plumes, which suggests that the evaporated primary organic aerosol can be precursors of SOAs in wildfire smoke plumes.

## 1 Introduction

As a result of fire suppression and climate change, wildfires in the western United States (U.S.) are becoming larger and more frequent, leading to deteriorating air quality (Dennison et al., 2014; Iglesias et al., 2022; McClure and Jaffe, 2018; Westerling, 2006). While CO and $CO_2$ dominates biomass burning (BB) emissions, organic compounds are more important in the context of air quality (Andreae and Merlet, 2001; Fine et al., 2004; Permar et al., 2021). Also, biomass burning is the main global source of primary carbonaceous aerosols and the second largest global source of non-methane organic compounds (Akagi et al., 2011; Bond et al., 2004). Many organic compounds directly emitted from biomass burning are hazardous air pollutants (Kim et al., 2018; O'Dell et al., 2020), and the atmospheric transformation of primary wildfire emissions can produce secondary organic aerosol (SOA) and ozone (Gong et al., 2017; Jaffe and Wigder, 2012; Liang et al., 2022; Lim et al., 2019), both of which have negative impacts on human health (Jerrett et al., 2009; Tuet et al., 2017; Wong et al., 2019). Understanding the chemical composition and transformations of biomass burning organic aerosol (BBOA) is therefore needed to predict the impact of wildfire smoke on human health and atmospheric chemistry.

The emissions from wildfires are critical inputs to atmospheric models used for assessing the effects of wildfire smoke. Although significant progress has been made in the chemical characterization of wildfire emissions in recent years, improved understanding of the emission profiles of photochemically reactive compounds, toxic compounds (such as polycyclic aromatic hydrocarbons, PAHs), intermediate-volatile and semi-volatile organic compounds (I/SVOCs) and primary organic aerosol (POA) are still needed (Andreae, 2019; Hatch et al., 2018). Speciated measurements are necessary because even compounds with the same molecular formula (e.g., monoterpenes) can have very different OH reactivities and SOA yields (Atkinson and Arey, 2003; Lee et al., 2006a, 2006b). In addition, as smoke dilutes, organic compounds in the particle phase can evaporate and can produce SOA when these I/SVOCs are then oxidized in the atmosphere (Robinson et al., 2007). Some studies have shown that dilution, evaporation and subsequent SOA formation in BB smoke plumes contributes more to SOA production than SOA formation from VOCs (Bruns et al., 2016; Grieshop et al., 2009; Palm et al., 2020). These conclusions were mainly based on bulk aerosol property measurements and/or subtracting the contribution from traditional gas-phase SOA precursors. Bulk measurements limit the ability to differentiate specific processes and sources, due to the lack of unique signatures (Zhang et al., 2018). Thus, identification and quantification of evaporated POA compounds are needed to better constrain the contribution of this process to SOA formation in BB plumes. Further, identification of particle-phase organic compounds at the molecular level can be useful for identifying marker compounds for source apportionment studies, and generally enable better understanding of SOA formation chemistry in wildfire plumes.




One-dimensional gas chromatography-mass spectrometry (GC-MS) has been widely used for molecular level measurements of biomass burning emissions in both the gas and particle phases. Many organic compounds in BB emissions have been identified and quantified by this technique (Fine et al., 2004; Hornbrook et al., 2011; Simoneit et al., 1993). This method is suitable for the characterization of compounds outstanding in the chromatogram, and for targeted quantification of known BB

tracers. However, it remains challenging to achieve full or near-full speciation of complex mixtures, such as BB emissions, due to the co-elution of many chemicals from the chromatographic column. It has been shown that unresolved compounds can contribute substantially to SOA formation (Jathar et al., 2014; Zhao et al., 2014). However, non-targeted molecular level measurements of organic emissions from biomass burning, particularly in the particle phase, remain scarce in the literature.

More complete characterization of complex mixture of gas- and particle-phase organic compounds is possible using two-dimensional gas chromatography (GC×GC), in which compounds are separated by both volatility and polarity. This method has been used to measure ambient OA composition in many urban and remote sites (An et al., 2021). Through GC×GC analysis, thousands of gas- and particle-phase compounds have been identified and quantified in BB emissions from lab combustion experiments (Hatch et al., 2015, 2018; Jen et al., 2019). Wildfire emissions, however, can be different from laboratory fires.

In wildfires, usually the fuel is a complex mixture of biomass from the crown to the understory and soil organic layer, burning at different stages. Many environmental factors such as fuel bed characteristics and meteorology can affect the combustion processes (Andreae, 2019; Ottmar, 2014). In August 2019, the Aerodyne Mobile Laboratory (AML), equipped with a comprehensive suite of real-time instruments (Herndon et al., 2005; Kolb et al., 2004; Yacovitch et al., 2019), traveled very close to three wildfires in the western U.S., as a part of the Fire Influence on Regional to Global Environments and Air Quality

(FIREX-AQ) campaign. In this work, we focus on the analysis of organic gases and particulate matter with diameters less than 2.5 µm ($PM_{2.5}$) collected using sorbent tubes and filters, respectively, on board the AML. Ground-based observations with the AML had the advantage of allowing sampling very close to the wildfires, which minimized the transformations occurring between emissions and measurements. The main objectives of this research were to (1) measure the emissions of gas- and particle-phase organic compounds from wildfires, explore the factors controlling the emissions, and compare the results with

the laboratory fires; and (2) investigate the effect of near-fire transformation of BBOA from the molecular perspective.

## 2 Materials and Methods

### 2.1 Fires, sampling routes

Smoke from four fires was sampled using the AML in this field campaign: the Nethker Fire near McCall, Idaho; Castle and Ikes Fires in Arizona; and 204 Cow Fire in Oregon. The sizes, fuel information, and canopy conditions of the fires are

summarized in Table 1. Although the sizes of the fires varied, fuel in all four fires was dominated by conifers. The perimeters of the fires and the sampling routes of the AML are displayed in Figure 1. The Castle and Ikes Fires were very close to each





other (Figure 1C). It is hard to separate the influences from the two fires especially for hourly samples, which were probably affected by emissions from both fires. We therefore consider the Castle and Ikes Fires together, as "Arizona Fires". As shown in Figure 1, some of the filter and sorbent tube samples were taken when the AML was stationary, while others were taken when the AML was in transit. More information about the fires, and operational details of the AML during this campaign can be found in Sumlin et al. (2021). We also collected $PM_{2.5}$ samples at a regional background site at the McCall Activity Barn (Longitude -116.115°, Latitude 44.872°, shown in Figure 1B).

### 2.2 $PM_{2.5}$ and VOC sampling by DEFCON

We collected 33 hourly samples (including 2 blanks and 5 background samples) on the AML during this campaign, using a custom-made sampler named DEFCON (Direct Emission Fire CON-centrator), which was mounted on the AML. A diagram of this sampler has been published in the supplement of Jen et al. (2019). Outside air was sampled from the front of the AML through a very short section of 3/8" (outer diameter) copper tubing. Air was subsampled at 150 ccm through a glass fiber filter coated with sodium thiosulfate (which removes $O_3$ to avoid oxidation artifacts), onto a dual-bed sorbent tube of Tenax TA (35/60) and Carbograph 1 (60/80), which collects the VOCs. The remaining sample flowed at 10 lpm through a $PM_{2.5}$ cyclone to remove large particles, and then through a 47 mm diameter circular quartz filter to collect $PM_{2.5}$. We encountered a leak problem on the VOC channel during the Nethker Fire sampling. Those samples were excluded from our analysis. We also collected and analyzed 33 $PM_{2.5}$ 3.5-hour samples (sampling flow rate: 44 lpm) on 102 mm diameter quartz filters using a sequential sampler (Yee et al., 2018) at McCall Activity Barn, from Aug 14 to Aug 28, 2019.

### 2.3 Filter analysis by GC×GC EI/VUV HRToFMS

The filters were stored at -20°C prior to analysis. Filters were analyzed using an offline GC×GC coupled to an electron impact/vacuum ultraviolet light ionization source and a high-resolution time-of-flight mass spectrometer (GC×GC EI/VUV HRToFMS), following the same protocols as Jen et al. (2019) and Liang et al. (2021). Small punches of each filter (with added isotopically labeled internal standards) were thermally desorbed at 320 °C in helium flow using a Gerstel Thermal Desorption System. The helium gas stream was saturated with N-Methyl-N-(trimethylsilyl) trifluoroacetamide (MSTFA) for online derivatization (which replaces the hydrogen in -OH, -SH and $-NH_2$ groups in molecules with a trimethylsilyl group) during thermal desorption. The compounds were trapped at 30 °C on a quartz wool glass liner prior to injection onto the first column. Compounds were first separated by volatility with an Rxi-5Sil MS column then by polarity with an Rtx-200 MS column (both from Restek). Electron impact (-70 eV) HR-ToFMS (Tofwerk, m/Δm ≈ 4000) was then used to ionize and detect the mass fragments of the separated compounds. In addition, vacuum ultraviolet light (VUV, 10.5 eV), a form of soft ionization provided by the Advanced Light Source at Lawrence Berkeley National Laboratory, was used in separate analyses of the same samples to provide parent mass information for individual organic compounds. Details about the columns and thermal programs are listed in Table 2. The chromatograms were analyzed using GC Image software (GC Image, LLC). Observed compounds were classified into aliphatic mono-carboxylic acid (monoacid hereafter), alcohol, alkane (plus a few minor alkenes), aromatic





(mono-cyclic only), nitrogen-containing (N-containing), sulfur-containing (S-containing), other oxygenated (non-aromatic and

with 2 or more -OH or -COOH groups), PAH (including substituted/oxygenated), sugar (and sugar derivatives including anhydro-sugars and sugar alcohols), terpenoid (including sesquiterpenoid, diterpenoid, triterpenoid and resin acid) and unknown groups. Compound identification and classification procedures, which involves matching with authentic standards, comparison with mass spectral libraries, and inference from parent ions, have been described in our previous work (Liang et al., 2021).

The compound quantification procedure, uncertainty and detection limits were also documented in Jen et al. (2019) and Liang et al. (2021). We upgraded the quantification method by adding more BB-related compounds in our standard mix (list can be found in the Supplement, Table S1). We injected multiple known concentrations of a 142-compound standard mix along with the internal standard mix to blank filters and obtained a response curve for each compound based on total ion count. Sample compounds within this standard mix were quantified using these curves. Sample compounds not in this standard mix were

quantified using the response curve of nearby (linear retention index difference < 200) standard compounds with high mass spectral cosine similarity (Isaacman-VanWertz et al., 2020; Stein and Scott, 1994), or the nearest compound on the GC×GC space if none of the nearby compounds have high mass spectral similarity with the compound being quantified. We quantified 240 compounds which includes the top 100 compounds (by signal) in three representative samples (one from each fire), as well as potential marker compounds for biomass burning. These compounds cover more than 75% of total chromatographic

signal of analytes, and include almost all the compounds we can identify (details in the Supplement Table S1). The subcooled saturation vapor pressure and therefore volatility distribution of particle phase organic compounds were also estimated, with details described in the Supplement. After that, the saturation vapor pressure $v_{P,i}$ of each compound was converted to the effective saturation mass concentration $C^*$ (in µg m⁻³) by:

$$C_i^* = \frac{\gamma_i v_{P,i} \mathrm{MW}_i}{RT} \tag{1}$$

where $\mathrm{MW}_i$ is the molecular weight of compound $i$ (before derivatization) in g mol⁻¹. We assume MW = 200 g mol⁻¹ for compounds with unknown formulae, following Isaacman et al. (2011). $\gamma_i$ is the unitless activity coefficient of compound $i$ (assumed to be 1), $v_{P,i}$ is in Pa, R is the gas constant (8.314 J mol⁻¹ K⁻¹) and T is the temperature (assume 298 K) (Isaacman et al., 2011; Pankow, 1994).

**2.4 VOC analysis by GC×GC ToF-MS**

The sorbent tubes were analyzed by another GC×GC ToF-MS instrument (Leco Corp., St. Joseph, MI), following the protocol described in Hatch et al. (2019). In brief, the samples were thermally desorbed at 285°C to a focusing trap, then injected to the GC×GC. The GC×GC comprises a DB-VRX primary column (Agilent, Santa Clara, CA) to separate compounds based on volatility and a Stabilwax secondary column (Restek, Bellefonte, PA) to separate compounds based on polarity. Details about the columns and thermal programs can be found in Table 2. Raw chromatograms were processed and analyzed using the





Chromatof software (Leco Corp., St. Joseph, MI). Compound identification and quantification procedures can be found in Hatch et al. (2019) and Hatch et al. (2015).

**2.5 Additional measurements**

Punches of filter samples collected from AML and McCall Activity Barn were sent to the Air Quality Research Center at UC Davis for Organic Carbon (EC) and Elemental Carbon (EC) analysis by a Sunset Model 5 Lab Carbon Aerosol Analyzer, following the NIOSH870 protocol. In addition to total OC and EC, this analysis also provides thermograms of OC and EC. On the AML, a soot particle aerosol mass spectrometer (SP-AMS) was used to measure the chemical composition of $PM_{2.5}$ (Fortner et al., 2018; Onasch et al., 2012). The SP-AMS switched between the conventional vaporization mode (standard mode) and the laser vaporization mode (SP mode, in which the conventional heater was also on), but was operating in the SP mode for the majority of time. For consistency, only data from the SP mode was used. The parameters $f_{CO2+}$ and $f_{C2H4O2+}$ (fraction of $CO_2^+$ and $C_2H_4O_2^+$ in the total organic signal, respectively) were determined. An Aerodyne Vocus PTR-ToF-MS (Krechmer et al., 2018) was also deployed on the AML for the for VOC measurements. Results from Vocus measurement, such as emission factors and comparison with GC×GC measurements will be reported elsewhere (Majluf et al., in prep). Concentrations of furan ($C_4H_4O$) and acetonitrile ($CH_3CN$) measured by Vocus were used in this study, as markers of short-lived BB VOC and stable BB VOC, which reflect the ages of BB plumes.


A Hemisphere GPS compass (Vector V103) was mounted on the AML to measure its real-time position. Wind speed and direction were measured by an RM Young Model 86000 3D anemometer, and were subsequently corrected for vehicle movements using the positioning data. CO was measured by a tunable infrared laser direct absorption spectrometer (TILDAS; Aerodyne Research Inc.); $CH_4$ was measured with a TILDAS $C_2H_6/CH_4$ instrument (Aerodyne Research Inc.) (McManus et al., 2015). $CO_2$ was measured by a Licor 6262 $CO_2/H_2O$ analyzer.

**2.6 Emission factor, emission ratio and modified combustion efficiency (MCE) calculations**

We calculated emission factors by the carbon mass balance method (Yokelson et al., 1999), using

$$\text{EF}_i(\text{g kg}^{-1}) = F_c \times 1000 \text{ (g kg}^{-1}) \times \frac{\text{MW}_i}{\text{MW}_c} \times \frac{\Delta X_i}{\Delta\text{CO} + \Delta\text{CO}_2 + \Delta\text{CH}_4} \quad (2).$$

In Equation 2, $\text{EF}_i$ is the emission factor of compound $i$ or particle-phase OC, EC. $F_c$ is the mass of carbon in the fuel, which varies between 0.45 and 0.55 for different vegetation (Burling et al., 2010). We assume $F_c = 0.5$ in this analysis. $\text{MW}_i$ is the molecular weight of compound $i$ in g $\text{mol}^{-1}$ (12 g $\text{mol}^{-1}$ for particle-phase OC and EC), and $\text{MW}_c$ is the atomic weight of carbon (12 g $\text{mol}^{-1}$). $\Delta X_i$ is the background-subtracted hourly integrated concentration of compound $i$ in moles per $\text{m}^3$, and $\Delta\text{CO}$, $\Delta\text{CO}_2$ and $\Delta\text{CH}_4$ are background-subtracted hourly integrated concentrations of CO, $CO_2$ and $CH_4$ in moles per $\text{m}^3$, respectively. To compare the current study with laboratory combustion studies, we also proposed a method to adjust the emission factor based



on the emission factor of CO (Supplement Section 3). Emission ratios (ERs) with respect to CO are determined by $\Delta X_i/\Delta CO$. The background level of CO in this field campaign varied between ~80 ppb and ~120 ppb.

Emission from biomass burning is strongly dependent on the combustion efficiency (Akagi et al., 2011; Jen et al., 2019; Yokelson et al., 1999). The modified combustion efficiency, MCE (Equation 3) is used to indicate the relative contribution

from flaming and smoldering combustion.

$$MCE = \frac{\Delta CO_2}{\Delta CO + \Delta CO_2} \quad (3).$$

Higher MCE indicates more complete combustion, in which more of the organic compounds are oxidized fully into $CO_2$. Flaming combustion has MCEs close to one, while smoldering combustion has MCEs between ~0.65-0.85 (Akagi et al., 2011). We encountered an interfering source (possibly ecosystem respiration, or multiple fires with different emission characteristics)

when calculating the $\Delta CO_2$ for some hourly samples. An example is shown in Figure S1. In such cases, $CO_2$ has relatively poor correlations with CO. To avoid this complication, for EF and MCE calculations, we also required the CO and $CO_2$ within the sampling hour to have an $R^2$ above 0.5. For both EF and ER, a threshold of sample average CO = 500 ppb was used to select samples predominantly affected by wildfire smoke, and to reduce the uncertainty associated with the background $CO_2$ estimation. Eleven filter out of 27 samples met all the thresholds for EF calculation.

**3 Results and Discussion**

**3.1 Organic particulate matter emitted from wildfires**

The emission of organic PM is strongly dependent on MCE. Figure 2 shows the EFs of particle-phase OC and EC as a function of MCE. The MCEs span from 0.86 (smoldering dominated) to 0.96 (flaming dominated). Consistent with lab combustion experiments (Jen et al., 2019), the EF of OC decreases with MCE. In Jen et al (2019), $EF_{OC}$ spanned two orders of magnitude

(1-100 g $kg^{-1}$), while the range of $EF_{OC}$ in the current study is within one order of magnitude, consistent with the similar fuel composition of the Nethker Fire and the Arizona Fires. However, in contrast to the lab fires in which $EF_{EC}$ increased as MCE increased (Jen et al., 2019), the $EF_{EC}$ has very weak correlation with MCE. The EC/OC ratio has no dependence on MCE either (Figure S2). This is partially because the fire plumes we encountered do not have very high MCE values. According to Pokhrel et al. (2016), the EC/OC ratios were almost constant when MCE was below 0.96-0.97, and then increased sharply at higher

MCEs. This further suggests that MCE is a useful parameter for predicting biomass burning emissions of particle-phase organic compounds as long as the fuel compositions of the fire whose EF is to be predicted is similar to the fire that the relationship was developed (e.g., both dominated by conifers).

The EFs of total quantified OA and most classes of organic compounds also have inversely proportional logarithmic

relationships with MCE (Figure 3). Data points from different fires appear close to the same line. The $R^2$ values of the fits are





higher than those reported in Jen et al. (2019), in which values were for a range of fuel species and components, likely because the fuels were more similar in the sampled wildfires than in the laboratory. Nitrogen-containing compounds do not have good correlation with MCE (not shown here) or $CH_3CN$. The nitroaromatic compounds are probably secondary, with some formation very close to the fires. The emissions of reduce nitrogen-containing compounds are dependent on the fuel nitrogen

content, which is not directly related to MCE (Coggon et al., 2016). for the EFs of individual compounds observed, 70% of them also depend on MCE with $R^2 > 0.5$, most of which with an inverse proportional relationship. Exceptions include 1,2,4-benzenetriol and 3-hydroxybenzoic acid, whose EFs increase with MCE. The emission factors, and emission ratios of all the compounds, are provided in the Supplementary spreadsheet.

The fits of EF vs. MCE for wildfires (this study) and the FIREX lab experiments (Jen et al. (2019)) are shown in Figure S3 for particle-phase OC, total quantified OA mass, aromatics, PAHs, sugars, and terpenoids+resin acids. Although for a given MCE the EF was lower for particle phase-OC and total quantified OA mass, markedly higher terpenoids+resin acids EF (mainly diterpenoids and resin acids) were measured from the wildfires. Terpenoids on average accounted for 36% (up to ~45% for two heavily loaded samples) of quantified particle-phase mass, which was comparable or higher than the contribution of sugar

and sugar derivatives. Dehydroabietic acid, didehydroabietic acid (6,8,11,13-abietatetraen-18-oic acid), isopimaric acid, pimaric acid and methyl dehydroabietate were among the top 10 compounds in terms of average EFs. The resin acids, which are located in resin ducts, are abundant in conifer stems and needles (Eksi et al., 2020; Krokene, 2015; Ramage et al., 2017; Simoneit et al., 1993).. The high contribution of terpenoids in the wildfire smoke samples confirms the dominance of conifers in the fuels burned in the fires we measured. The high emission of diterpenoids and resin acids in wildfires compared with lab

combustion likely came from heat-induced evaporative emissions from non-burned forest components in the wildfires. Melting points of these resin acids are between 150 ℃ and 200 ℃, and their 50% loss temperature in thermogravimetric analysis are around 270℃-300℃ (Schuller and Conrad, 1966). Peak temperatures in flaming and smoldering combustion are 1500-1800℃ and 450-750℃, respectively (Santoso et al., 2019). Heat from the fires likely drove melting and evaporation of resins releasing the observed terpenoids and resin acids from biomass that was not burned, and it seems this source represented a substantial

fraction of the total emissions. In the Fire Lab 2016 burns of coniferous ecosystem fuel components, the contribution of terpenoids to total quantified OA mass was always lower at 2-20% (Jen et al., 2019). However, in that study, the EFs are for the fire-integrated samples, which probably have lower evaporative emissions from unburned fuels. The evaporation of volatile compounds after emission but before the smoke was sampled probably played a less important role in causing the high mass fraction of terpenoids in the wildfire samples. We modeled the particle phase fraction of compounds as smoke dilutes from

OA = 400 µg m⁻³ to 80 µg m⁻³ (close to the OA of the least concentrated sample used in EF calculation), assuming equilibrium partitioning. The evaporated fraction of sugar caused by this dilution process is estimated to be around 15%, which cannot cause a large increase in the relative abundance of terpenoids. Also, there is not a clear relationship between terpenoid/total quantified OA vs. OC (Figure S11).



Based the slopes of the fits of EFs vs MCE, in both lab combustion and wildfires, the EFs of PAHs have the strongest dependence on MCE among all the chemical classes (largest slope). PAHs on average represent 7.6% of total quantified OA by EF in this campaign. PAHs are high-temperature pyrolysis products from biomass burning (Collard and Blin, 2014; Sekimoto et al., 2018), or thermal decomposition products of diterpenoids (including resin acids), such as retene (Ramdahl, 1983; Simoneit et al., 1993; Standley and Simoneit, 1994). Most PAHs we observed are probably thermal decomposition

products of diterpenoids, as their structures suggest.

The volatility distribution of observed particle-phase organic compounds in three samples is shown in Figure 4 (in effective saturation mass concentration $C^*$ space) and Figure S4 (in deuterated-alkane retention index space). The samples represent the most heavily loaded (highest OA mass concentration) sample collected near each fire. The fraction of a compound $i$ in the

particle phase ($F_{p,i}$) can be calculated by

$$F_{p,i} = \frac{C_{OA}}{C_{OA} + C_i^*} \qquad (4)$$

where $C_{OA}$ is the total concentration of organic aerosol in μg m$^{-3}$ (Donahue et al., 2006). When $C^*$ for a compound is equal to $C_{OA}$, 50% of this compound will be in the gas phase and 50% will be in the particle phase. Sugars and sugar derivatives such as levoglucosan have effective saturation vapor pressures between 10-100 μg m$^{-3}$. Thus, when smoke plumes dilute to those

levels, some sugars can evaporate and react with oxidants. That makes them less ideal tracers for BB emission in source apportionment studies (Hennigan et al., 2010). On the contrary, as Figure 4 shows, the diterpenoids and resin acids have effective saturation mass concentrations between 0.1-1 μg m$^{-3}$, which means they are much less prone to evaporation. Given the high abundance of these compounds in conifer fires, we believe these compounds can be useful tracers for biomass burning. Furthermore, diterpenoids and resin acids have lower oxygen to carbon ratios compared with sugars and sugar derivatives. If

they eventually evaporate and get oxidized by OH in the gas phase, they are less likely to fragment like sugars (Donahue et al., 2013). Instead, ketones and -OH groups will be preferentially added, which will lower the saturation vapor pressure of the original compound, and lead to net SOA formation in the remote atmosphere. That can shift the volatility distribution to the less volatile end.

**3.2 VOC emission from wildfires**

The summed EFs for 13 different VOC classes are shown in Figure 5. EFs and ERs of individual VOCs are given in the data Supplement spreadsheet. Furanoids and monoterpenes have the highest EFs among the measured VOCs. The monoterpene EFs, particularly in the Nethker Fire, are higher than those measured in smoke samples from coniferous species burned during the Fire Lab 2016 study (Hatch et al., 2019), with the exception of the canopy samples (Figures S5). They are also higher than those measured in smoke samples from a mixed conifer prescribed burn in Blodgett Forest (Hatch et al., 2019). The field

adjusted EFs for monoterpenes in the Nethker Fire is also higher than most of the samples in the Fire Lab 2016 study (Figure



S6). The high emission of monoterpenes, in addition to high emission of diterpenoids and resin acids in the particle phase, can be attributed to the distillation of these compounds from plant resin ducts (Hatch et al., 2019; Koss et al., 2018). Lab combustion experiments shows such distillation processes happen in the early phase of fires (Koss et al., 2018; Sekimoto et al., 2018). The ubiquity of monoterpenes in our samples collected on different dates indicates the fire fronts probably

approached some new fuel continuously. Limited by the number of samples that met the criteria outlined in Section 2.6, we were not able to assess the relationship between the VOC EFs and MCE. The ERs of the observed VOCs and terpenes are shown in Figures S7 and S8. The ERs of furanoids and monoterpenes are higher than in the Fire Lab 2016 study. Alpha-pinene, 3-carene, camphene, beta-phellandrene, beta-pinene and limonene are the most abundant monoterpene species observed in this study. This is very similar to the speciation observed in the prescribed burns in Blodgett Forest, which is consistent with the

high similarity of fuels in the two field campaigns. The similarity between the fuels in the two campaigns is also supported by the similar benzonitrile/furfural emission ratios. The log-transformed benzonitrile/furfural ratio has been demonstrated to be a good indicator for different plant tissues (Coggon et al., 2016; Hatch et al., 2019). The log-transformed benzonitrile/furfural emission ratio from the wildfires in this study is very close to the ratios of the Blodgett burns. These ratios fall between burning woody fuels and composite fuels (included all individual fuel components) in the Fire Lab 2016 study, indicating the

dominance of wood tissue combustion in the wildfires in this study (Figure 6).

There are some differences of monoterpene emissions among the three fires. In the 204 Cow fire, 3-carene is the dominant isomer of monoterpenes, while it is less abundant in the Nethker Fire (Figure S8). It was reported that 3-carene emission from burning wood is significantly higher than burning needles of black spruce and ponderosa pine (Hatch et al., 2015). Since the

more definitive acetonitrile/furfural ratio analysis shows the dominance of woody fuels in all three fires, there are probably factors other than the needle/wood fraction caused the difference of 3-carene emissions. $\beta$-Phellandrene, which dominates the emissions from lodgepole pines (Hatch et al., 2019), contributes 16% and 8% to total monoterpene emitted in the Nethker Fire and the 204 Cow Fire, respectively. It only accounts for 2% of monoterpene emissions from the Arizona Fires, which agrees with the presence of lodgepole pine as a fuel in the Nethker and 204 Cow Fires, and the absence of lodgepole pine in the

perimeter of the Arizona Fires. In Figure 7, we examined the emission profiles of compounds representative of the fuel types, found in Stamatis and Barsanti (2022). Figure 7 suggests that the low abundance of 3-carene in the Nethker Fire emission might be due to the absence of ponderosa pine in that region. Bornyl acetate, the tracer for Douglas-fir and subalpine fire smoke (Hatch et al., 2019), accounts for 3% of terpene emissions in the Nethker Fire, but only 0.67% and 0.13% of terpene emissions in the Arizona Fires and the 204 Cow Fire, respectively. This difference suggests that a higher fraction of fuel burned

in the Nethker Fire when we took the samples was Douglas-fir or subalpine fir. However, the abundances of bornyl acetate in the emissions from these wildfires are still much smaller than the source signature of burning Douglas-fir or subalpine fir, which were important plants present in the Nethker Fire and the Castle Fire's perimeters. This discrepancy might be caused by the inhomogeneity of fuel distribution, and these fir species were not burned when we took the samples.



## 3.3 Near fire transformation of particle-phase BBOA

### 3.3.1 Nitrogen-containing compounds and oxygenated compounds

Nitrogen-containing compounds and oxygenated compounds (such as oxygenated aromatic compounds and multifunctional acids) are common markers for aged BBOA (Bertrand et al., 2018; Liang et al., 2021). In the samples collected in this campaign, nitrogen-containing compounds (mainly nitroaromatic compounds) only represent 2.1% of total quantified OA on average. These compounds were observed from the Fire Lab fresh emission, but they are mainly SOA compounds formed when aromatic compounds react with OH, $NO_3$ and HONO in the presence of $NO_2$ (Bertrand et al., 2018). Since our samples were collected close to fires, the concentration and contribution of nitrogen-containing compounds to total quantified OA were lower than that observed in BBOA measured 50-60 km downwind from northern CA wildfires (Liang et al., 2021). The average nighttime concentration of nitroaromatic compounds were higher (but not statistically significantly higher) than in the daytime samples, possibly due to the higher yield of $NO_3$ oxidation (Finewax et al., 2018), which is in agreement with the airborne plume study in FIREX-AQ (Decker et al., 2021). Oxygenated compounds can either be directly emitted or be produced during atmospheric oxidation. The oxygenated compounds have low concentration in most samples, on average representing less than 1% of total quantified particle-phase OC. As shown in Figure 3, the EFs of this group of compounds have similar dependence on MCE with other groups of compounds, which indicates these oxygenated compounds observed near the fires are mainly primary emission, instead of atmospheric aging products. We observed possible SOA marker compounds such as butanedioic acid and octanedioic acid in the samples. A later-generation day-time oxidation product of BBOA that is typically observed in aged smoke, malic acid, was not detected in any of the samples (including the background samples).

### 3.3.2 Molecular evidence of primary BBOA evaporation

We tracked the behaviors of individual particle phase compounds. It is hard to estimate the physical plume age for each sample because the AML was moving when we took most of the samples, due to variable meteorological conditions and fire heterogeneity, particularly in the near field. Instead, we used ln(Acetonitrile/Furan) as an metric for photochemical plume age. Furan is a short-lived primary BB VOC, with a lifetime of ~4 hours against OH oxidation and ~1 hour against $NO_3$ oxidation under typical troposphere conditions (OH = $1.5 \times 10^6$ molecule $cm^{-3}$, $NO_3$ = $2.5 \times 10^8$ molecule $cm^{-3}$) (Kind et al., 1996), while acetonitrile is an inert tracer for biomass burning emissions (Atkinson et al., 2006; Holzinger et al., 1999). It can be expected that ln(Acetonitrile/Furan) will increase with plume age, because furan is quickly oxidized in the atmosphere. We chose to use ln(Acetonitrile/Furan) as the main metric here because of the continuous high time resolution data available from the Vocus measurements and the ability to compare with simultaneous SP-AMS measurements. In SP-AMS mass spectra, fragment $CO_2^+$ is a marker for oxygenated organic aerosol while the fragment $C_2H_4O_2^+$ is a marker for anhydrous sugars such as levoglucosan (Alfarra et al., 2007; Collier et al., 2016; Hodshire et al., 2019). As shown in Figure S9, ln(Acetonitrile/Furan) has strong positive correlations with both $f_{CO2+}$ (fraction of $CO_2^+$ in the total organic signal) and $f_{CO2+}/f_{C2H4O2+}$. This supports the use of ln(Acetonitrile/Furan) as the metric of plume age. Figure 8 shows that the fraction of summed concentration of





observed particle phase compounds with effective saturation mass concentration $C^* < 1$ µg m$^{-3}$ (extremely low-volatility organic compounds and low-volatility organic compounds) increased with ln(Acetonitrile/Furan). The same trend is observed when $f_{CO2+}$ and $f_{CO2+}/f_{C2H4O2+}$ are used as plume age metrics, though there are fewer data points (Figure S10). This is probably caused by the evaporation of compounds with $C^* \geq 1$ µg m$^{-3}$ (SVOCs and IVOCs), and the formation of SOA compounds not

measured by GC×GC. We calculated the correlation coefficient ($r$) between the mass fraction of each compound in total quantified OA and $f_{CO2+}$. If there was not any variation in the initial emission for each sample, for each compound, if $r > 0$ (equivalent to a positive slope for the linear fit), its mass fraction in total quantified increased in aging. If $r < 0$, its fraction decreased in aging. The two most abundant primary biomass burning compounds (on average account for 30% of total quantified OA in Nethker Fire) have $r$'s with different signs. Levoglucosan, which has a $C^* = 10^{1.22}$ µg m$^{-3}$, has an $r$ of -0.33.

In contrast, dehydroabietic acid ($C^* = 10^{-0.11}$ µg m$^{-3}$), has an $r$ of 0.79. This suggests levoglucosan evaporated during aging. Dehydroabietic acid was probably not formed in aging, but its contribution to total quantified OA increased as the more volatile compounds evaporated. 4-Nitrocatechol, 3-methyl-5-nitrocatechol and 4-methyl-5-nitrocatechol, though have $C^*$'s near $10^3$ µg m$^{-3}$, have $r$'s of 0.85, 0.77 and 0.79. This is consistent with secondary production of the nitrochatechols (Bertrand et al., 2018).

### 3.3.3 Evolution of particle-phase semi-volatile OC from wildfires (OC1)

In the Fire Lab 2016 study (Jen et al., 2019), the Napa Fire 2017 study (Liang et al., 2021) and the FIREX-AQ field campaigns in 2019 (this study), we collected biomass burning-related PM$_{2.5}$ and PM$_1$ (Fire Lab 2016 only) filter samples from different environments and analyzed them for their particle phase OC and EC following the NIOSH 870 protocol. In this protocol, to determine the concentration of OC, each sample was heated to 310°C in helium for OC1, 475°C to get OC2, 615°C for OC3

and 870°C for OC4, and followed by a charring correction (Bautista et al., 2015; Wu et al., 2016). The OC1 can be seen as the semi-volatile fraction of OC. For filters collected on the AML, the fraction of OC1 in OC has negative correlations with both ln(Acetonitrile/Furan) and $f_{44}$ (coincidentally $r = -0.85$ for both), which suggests OC1 was consistently evaporated in the aging process. The fractions of OC1 in OC for samples across the four campaigns are shown in Figure 9. We found that the AML samples have the highest fraction of OC1, and are therefore more volatile. The Fire Lab 2016 samples have slightly lower

(though not statistically significantly lower than the AML samples) contribution of OC1 to total OC, which is probably related to the combustion process or the fuel (e.g., less evaporation of semi-volatile organic from unburned fuels in laboratory combustion). In samples collected farther away from the wildfires (McCall and Napa), the fractions of OC1 are lower. These observations demonstrate that as the smoke diluted, the IVOCs and SVOCs evaporated from the particle phase (Robinson et al., 2007). The SOA produced from these vapors should be less volatile than the reactants, which potentially contributed to

less volatile fractions of OC, consistent with our speciated measurements (Section 3.3.2).

## 4 Conclusions

In this work, we analyzed gas- and particle-phase organic compounds emitted from western US wildfires using GC×GC, and assessed the gas-particle partitioning behavior of these compounds after emissions, particularly as relevant for SOA formation. Emission factors (EFs) and emission ratios (EFs) were calculated and reported for 72 gas-phase and 240 particle-phase
compounds. In agreement with prior laboratory biomass burning studies, we found that MCE was a good predictor of particle-phase organic compound EFs, except for EC. Diterpenoids and resin acids dominated the particle-phase organic compounds quantified in the wildfire smoke samples, likely due to heat-driven evaporation. Similarly, monoterpenes in the gas phase were higher in the wildfire smoke samples than in most Fire Lab 2016 smoke samples. We confirmed that the evaporation of SVOCs took place in smoke plumes from the molecular perspective. Such evaporation led to a decrease of the semi-volatile fraction
of particle-phase OC when smoke transported downwind and diluted. The emission profiles reported here can benefit future source apportionment or modeling studies and exposure assessments. They can also help researchers focused on public health to identify compounds for more targeted studies to better understand the health impacts of wildfire smoke.

### Data availability

Emission factors and ratios of all the compounds are provided in the supplementary spreadsheet. Data measured by the AMS,
VOCUS and OC/EC data can be accessed at https://www-air.larc.nasa.gov/missions/firex-aq/index.html. Data used in this research are also available from the authors upon request.

### Author contribution

AHG, KCB, and SCH designed research, YL, CS, ECF, RAW, PVR, FM, TIY, CD and SCH performed research. YL, NMK
and AHG devised the sampling plan. NMK contributed research instruments. YL, CS, ECF and AHG analyzed data. YL, KCB and AHG wrote the manuscript with input from all coauthors.

### Competing interests

The authors declare that they have no conflict of interest.

### Acknowledgements

The authors acknowledge Rob Roscioli and Jordan Krechmer of Aerodyne Research, Inc. for their effort in contributing the AML data. The authors thank Robert Weber, Emily Franklin, Caleb Arata of UC Berkeley; Oleg Kostko, Bruce Rude and



Kevin Wilson of Lawrence Berkeley National Laboratory for their help during the beamline campaign. We also thank Coty Jen of Carnegie Mellon University for contributing filters from the Fire Lab 2016 study, Emily Franklin for contributing standard solutions. We thank Dennis Baldocchi of UC Berkeley for helpful suggestions. This research used resources of the
Advanced Light Source, which is a DOE Office of Science User Facility under contract no. DE-AC02-05CH11231.

**Financial support**

This work was supported by the National Oceanic and Atmospheric Administration (grant numbers: NA16OAR4310107, NA16OAR4310103, NA16OAR4310104) to UCB, UCR, and Aerodyne Research Inc., respectively.

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





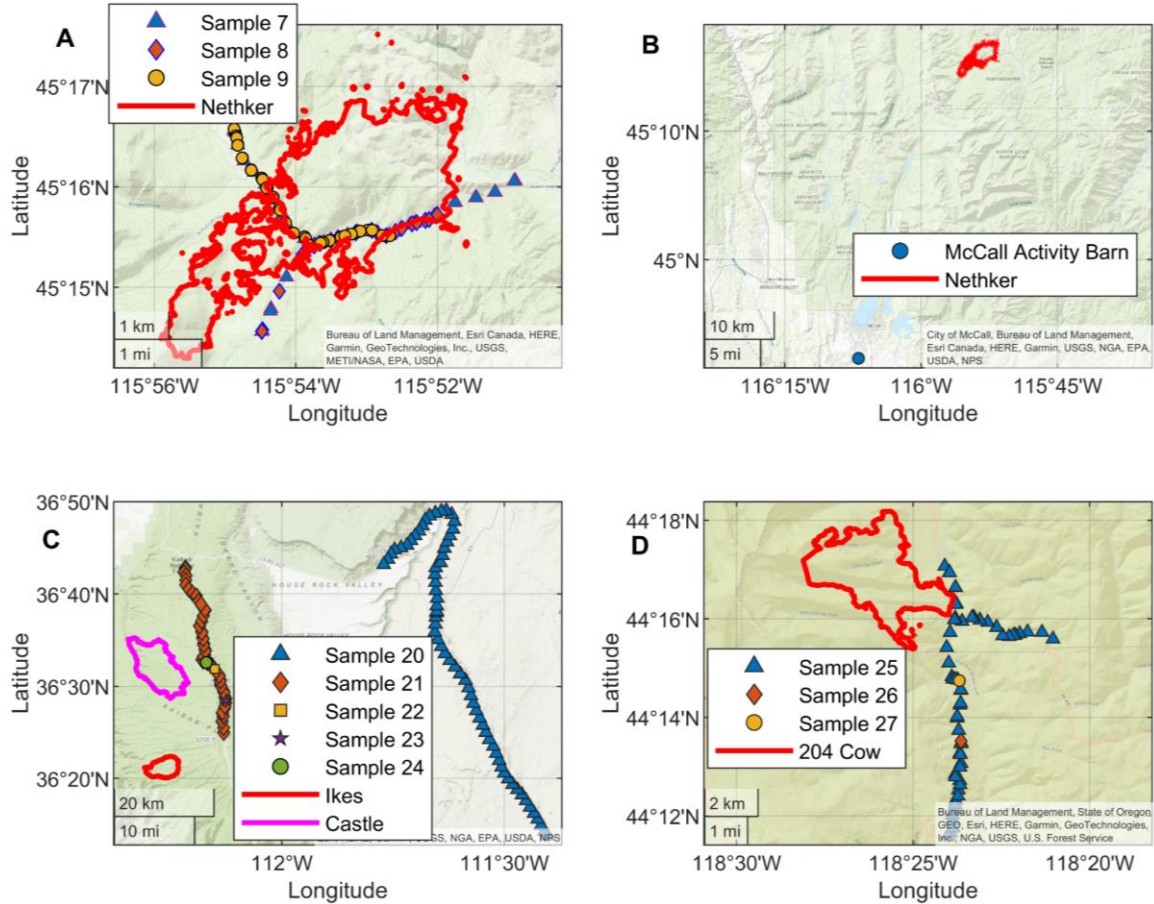

**Figure 1:** Locations of the AML (every minute) when samples were collected and perimeters of the fires (A & B. Nethker Fires, C. Arizona Fires, D. 204 Cow Fire). The AML made multiple trips to the Nethker Fire. Only three samples are shown here as examples. Fire perimeters shown here are the perimeters on or closest to the date when the samples were collected.





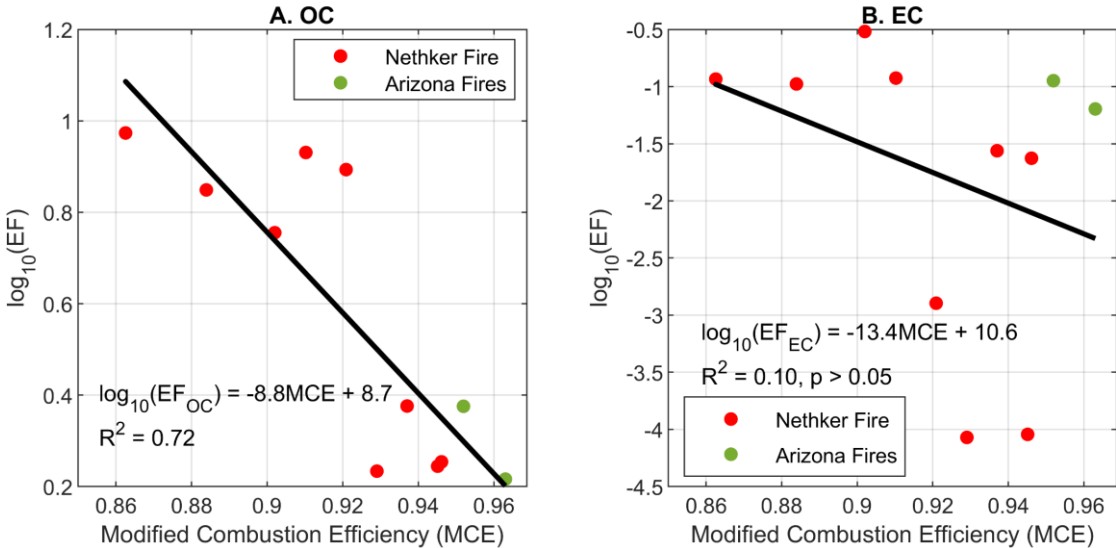

**Figure 2: (A)** Emission factors of OC as a function of modified combustion efficiency (MCE) and **(B)** EC vs MCE. Colors indicate the
670 different fires. Black lines indicate the log linear fits.



**Figure 3:** Summed emission factors for chemical classes as a function of MCE. Colors indicate the different fires. Black lines indicate the log linear fits. Compounds in each class are given in the supplementary spreadsheet.

675





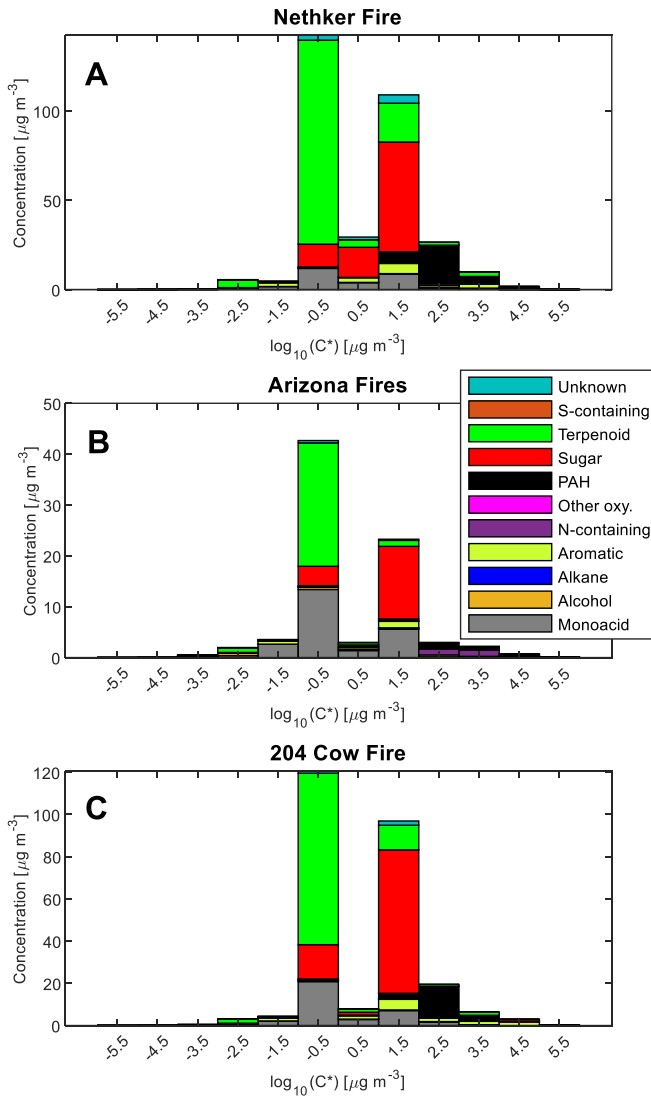

**Figure 4:** Mass distribution of speciated compounds in different effective saturation concentration bins. As an example, the bin centered at -0.5 includes compounds with $\log_{10}(C^*)$ between -1 and 1 µg m$^{-3}$. (**A**) is for the most heavily loaded sample collected near the Nethker Fire (OC = 352 µg m$^{-3}$); (**B**) is for the most heavily loaded sample collected near the Arizona Ikes and Castle Fires (OC = 138 µg m$^{-3}$); (**C**) is for the most heavily loaded sample collected near the 204 Cow Fire (OC = 260 µg m$^{-3}$). N = 1 for each plot.





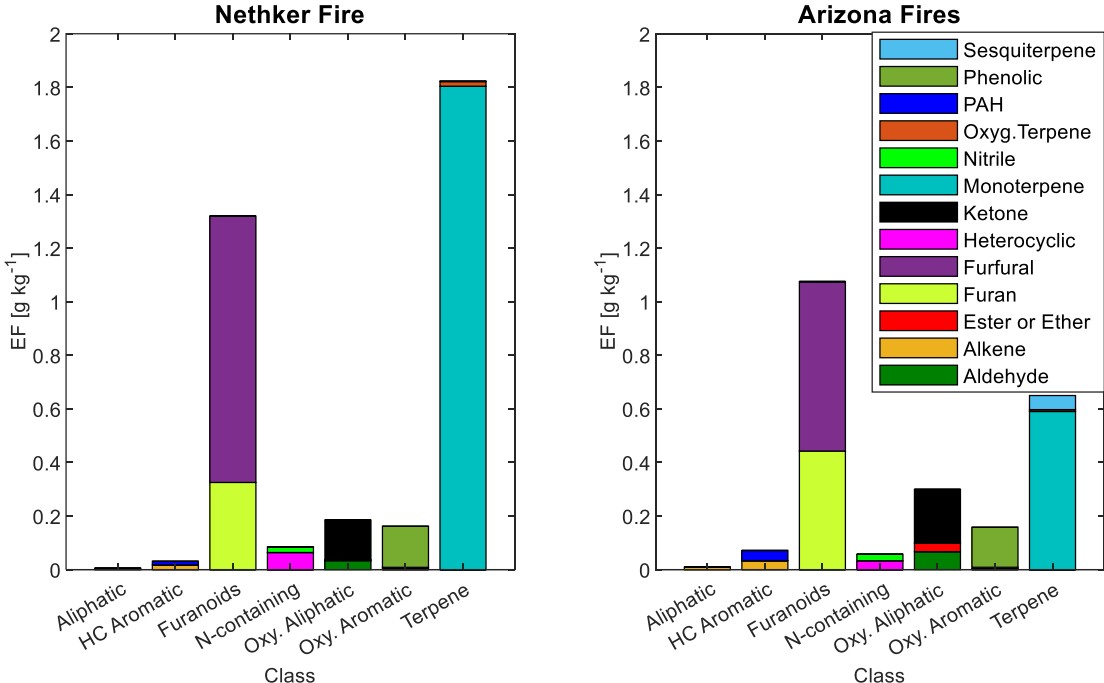

**Figure 5:** Emission factors of observed VOCs from the Nethker Fire (N = 1) and Arizona Fires (N = 2, averaged), grouped by chemical classification.

690

**Figure 6.** Swarm plots of the log-transformed ratio of biomass-burning tracers benzonitrile and furfural for the Fire Lab 2016 laboratory (Hatch et al. 2019) and this study.




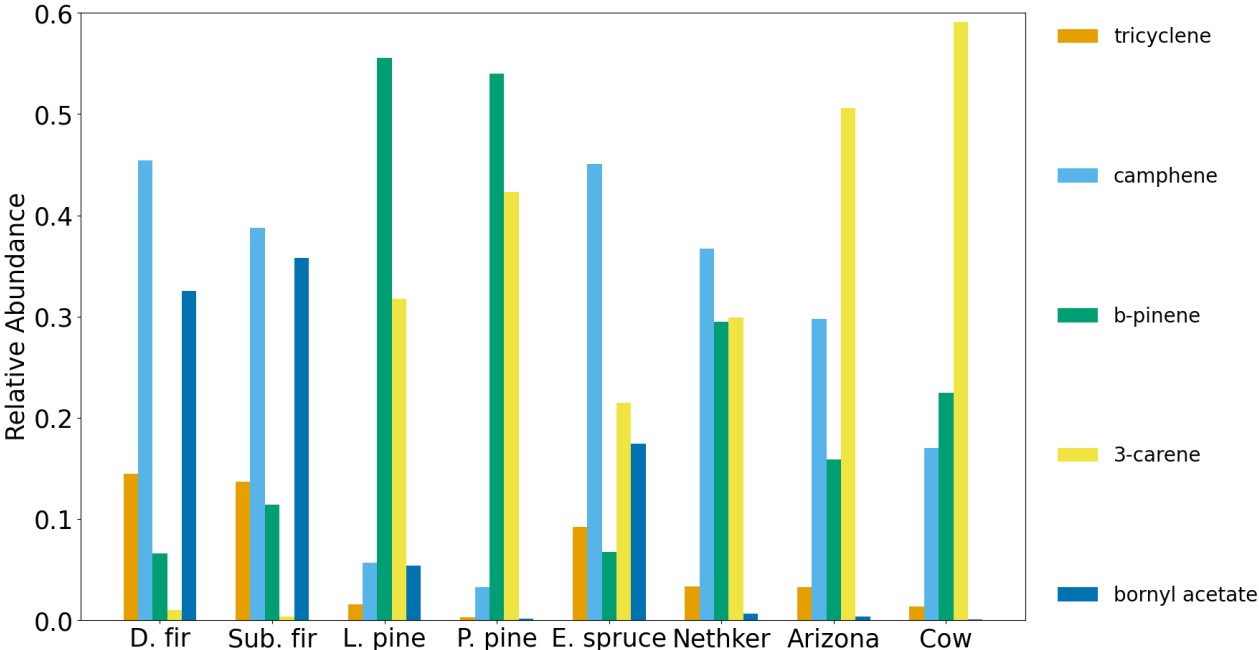

**Figure 7.** Relative abundance of emission ratios for five selected terpenoids in the Fire Lab 2016 study compared to wildfires in this work.





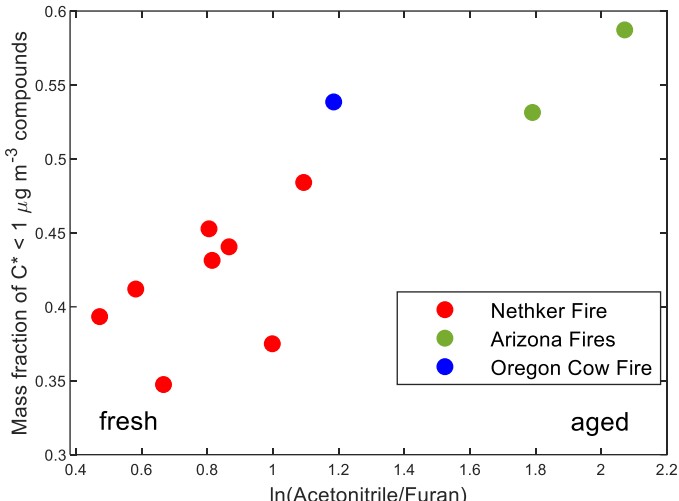

**Figure 8: A.** Mass fraction of summed concentration of observed particle phase compounds with $C* < 1$ µg m$^{-3}$ in total quantified OA, as
a function of photochemical age represented by ln(Acetonitrile/Furan).





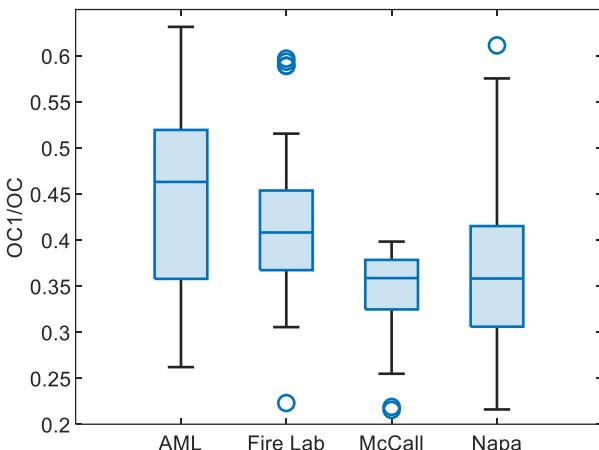

**Figure 9:** Box plot of the fraction of OC1 in OC for wildfire emissions. Each box plot shows the interquartile range with whiskers extended to 1.5 times the interquartile range. Central marks are the medians. Circles denote outliers.

705





**Table 1: Key characteristics of wildfires studied**

| Fire Name | Final Size (km²)ᵃ | Start Date | Fuels | Order/Classᵇ | State |
|---|---|---|---|---|---|
| **Nethker Fire** | 9.6 | Aug 04, 2019 | Lodgepole pine, Engelmann spruce, subalpine fir, big sagebrush | Tree dominated, close tree canopy | Idaho |
| **Castle Fire** | 78.4 | July 12, 2019 | Mixed conifer (e.g., ponderosa pine, Douglas fir) but also dead and downed trees | Tree dominated, open tree canopy | Arizona |
| **Ikes Fire** | 66.4 | July 25, 2019 | Mixed conifer, grass, and understory | Tree dominated, open tree canopy | Arizona |
| **204 Cow Fire** | 39.1 | Aug 09, 2019 | Lodgepole pine, mixed conifer, and dead and downed wood | Tree dominated, close tree canopy | Oregon |

a. The final sizes and fuels of fires are accessible at the Incident Information System's website (https://inciweb.nwcg.gov/).

710 b. The order/class information are taken from the LANDFIRE Existing Vegetation Type database (https://landfire.gov/evt.php).





**Table 2: GC×GC Materials and Methods**

|  | GC×GC - particle phase measurement | GC×GC - gas phase measurement |
|---|---|---|
| Thermal Desorption Unit | Gerstel TDS-3, TDSA2 | TurboMatrix 650, Perkin Elmer |
| Thermal Desorption Temperature | 320 °C | 300 °C |
| Column I (volatility) | Restek Rxi-5Sil-MS (60m, 0.25mm i.d., 0.25 µm film thickness) | Agilent DB-VRX (30 m, 0.25 mm i.d., 0.25 µm film thickness) |
| Column II (polarity) | Restek Rtx-200MS (1m, 0.25mm i.d., 0.25 µm film thickness) | Restek Stabilwax (1.5 m, 0.25 mm i.d., 0.5 µm film thickness) |
| Primary Oven Temperature Program | 40 to 320°C at 3.5°C/min, then hold for 5 min at 320°C. | At 35 °C for 5 min followed by a 2.5 °C/min ramp to 165°C, and 10 °C/min to 210 °C with a final hold of 1 min. |
| Secondary Oven Temperature Program | 15 °C higher than primary oven | 10 °C higher than primary oven |
| Mass Spectrometer | HR ToF-MS | ToF-MS |