# Peer review of "Emissions of Organic Compounds from Western US Wildfires and Their Near Fire Transformations"

_Atmospheric Chemistry and Physics, 2022_

## Author Comment (AC1)

Dear Editor and Reviewers,

Thank you for your comments and suggestions, which have served to improve the clarity and quality of this manuscript. Our point-by-point responses to the comments are in blue, and modifications to the text are in *italic brown*. Line numbers refer to the version of the manuscript showing tracked changes. Apart from the comments from the reviewers, we found an error in calculating the emission factors of OC and EC, which is fixed in this version (Figures 2, S3 and S11 are affected). The conclusions are not affected. We also modified Figures 2, 3, 5, 8 and Figures S2, S7, S9 and S10 to better allow readers with color vision deficiencies to correctly interpret our findings.

**Reviewer 1**

**Summary:**

The work submitted by Liang and Stamatis et al., addresses gas and particle phase organic species, and investigates the composition of PM from largely coniferous fuel wildfires sampled by the Aerodyne Mobile Lab as part of the FIREX-AQ study. The results presented here ultimately aim to address unknowns associated with the composition of wildfire PM, and the formation of secondary species such as SOA. Overall the manuscript is well written with minimal grammatical errors and the work is clearly presented. It is appropriate for ACP and should be considered for publication, but some minor caveats need to be considered first. Though the overall study is clear, the manuscript would benefit from increased specificity, especially in qualifying comparative statements or defining the importance of contributions from single species (see below in more detail). This would help provide additional context to some of the comparisons with other studies and to which compounds are most important in this context of wildfire PM. Additionally, though

the importance of this work is implied throughout the text, more clearly defined statements regarding broader impacts would help strengthen the significance of the results presented here. For instance, estimates on SOA formation or potential yields from the most abundant compounds could help contextualize the importance of these species, though a detailed analysis may be outside the scope of this work. Additionally, in the conclusion, the authors mention that the emission profiles can benefit future source apportionment and exposure assessments for public health and impacts. Though this is true, specificity on how that could be done is lacking. The authors should not underestimate the importance of the work they have done--they even mention that non-targeted molecular level measurements of biomass burning particles in the literature are scarce. Thus, the work is unique and the authors should reiterate that and the implications moving forward.

We appreciate the very insightful and detailed comments from Reviewer 1. We believe the quality of our analysis is greatly improved thanks to Reviewer 1's suggestions. Estimation of SOA yield of the emissions may have large uncertainties because the SOA yield of many IVOCs and SVOCs emitted in biomass burning are unknown, while these compounds were shown to be the major precursors of BB SOA. Nevertheless, we tried to strengthen our argument through discussion. Our point-by-point responses and text revisions are below.

**Abstract:** Could benefit from more specificity. E.g, "MCE was a good predictor of particle phase EFs…" is there an $R^2$ value or correlation coefficient range that can be listed to strengthen this statement?

We thank the reviewer for this suggestion. As shown in Figures 2 and 3, the $R^2$ values for different groups of compounds vary substantially. We would like to mention the $R^2$ values for the two dominant classes of compounds here. The text now reads:

*"As has been demonstrated previously, modified combustion efficiency (MCE) was a good predictor of particle phase EFs (e.g., $R^2$ = 0.78 and 0.84 for sugars and terpenoids, respectively), except for elemental carbon."*

**Introduction:**

L39, 58 & throughout: references within text should be in some chronological order, descending or ascending based on year

We thank the reviewer for the suggestion. The in-text citations are changed accordingly.

L83-84: consider rewording "traveled very close to three wildfires" to "sampled ground-based near-source emissions from wildfires"

It is changed as suggested.

L87: "sampling very close" quantify w/ distance (i.e <1 km)

We changed the sentence into:

*"Ground-based observations with the AML had the advantage of allowing sampling very close to the wildfires (sometimes even less than 100 m), which minimized the transformations occurring between emissions and measurements."*

L90: End of this paragraph would benefit from expected outcomes of tasks 1 &2. What do you expect these objectives to bring to the broader community/current knowledge? Reiterate points from intro. 1-2 sentences is fine.

We thank the reviewer for this suggestion. We would like to add the following sentence here.

*"The results from this work will be important inputs in atmospheric chemistry models, and can potentially give public health scientists targets for toxicological studies to better understand the health impacts of wildfire smoke."*

**Materials & Methods:**

L99-100: As samples were taken when the AML was stationary and mobile, was any process considered to remove potential cross-contamination from the AML? Was this a concern? If not, express why.

Cross-contamination from AML's exhaust for mobile samples is expected to be low because the air was drawn from the front of the vehicle. Also, as mentioned in Sumlin et al. (2021), when the AML did stationary sampling, it was parked with the sample inlet on the front of the truck, facing into the wind to avoid self-sampling of its own exhaust.

We added the following sentence to this paragraph:

*"To avoid self-sampling the exhaust of the AML, the ambient air was sampled from the front of the AML. During stationary sampling, the AML was parked with the sample inlet facing into the wind."*

L105: 'Very short section' – any quantifiable length?

The exact length was not measured, but based on the instrument setup in the mobile lab, the approximate length of the tubing was ~1.5 m. This information is added into the manuscript.

L110: Wouldn't necessarily call $PM_{2.5}$ large. The $PM_{2.5}$ cyclone was there to remove particles below $PM_{2.5}$.

We changed it to "*to remove particles with diameters larger than 2.5 μm*".

L111: Mention of samples excluded from analysis. What % is this compared to rest of data?

Most VOC samples were excluded because of the possible leak. Only 12 gas phase samples were reported in this work. We modified this sentence to:

"We encountered a potential leak problem on the VOC channel during the Nethker Fire sampling. *Those samples were excluded from our analysis, leaving results from 12 gas-phase samples reported in this work.*"

L112: suggest rewording "analyzed 33 $PM_{2.5}$ 3.5-hour samples" to "analyzed 33 3.5 hour $PM_{2.5}$…", less awkward

It is changed as suggested.

L156: Is 285C enough to desorb all species? Any fraction expected to be lost?

Sorry, it is an error here. It should be 300 ℃, as written in Table 2. This temperature is high enough for VOCs. Even heating to 285 ℃ allows 15-carbon compounds like sesquiterpenes to desorb from the collector (Hatch et al., 2015).

L204: 'Eleven filter out of 27 samples met all the thresholds for EF calculation' – awkward wording, consider rewriting

We would like to change it to:

*"Only 11 filter samples met all the criteria to be included in the EF calculation."*

**Results & Discussion:**

L215-217: Seems like you can condense this into "MCE is useful for predicting emissions among similar fuel types?"

We think the condensed version the reviewer proposed is great. The sentence is shortened as below.

*"This further suggests that MCE is useful for predicting emissions among similar fuel types."*

L270-275: Earlier authors mentioned that resin acids and diterpenoid species are abundant in the conifer stems and needles, but also note that abundance can come from heat-induced evaporative emissions from "non-burned forest components in wildfires". In this line, authors mention that these compounds can be used as tracers for BB. Given the earlier statement, with what certainty/accuracy can these species be used as tracers if there is still some fraction of them emitted in the presence of non-BB (heat induced)? Perhaps I am misinterpreting the results, but clarification on these two statements/stipulations in regards to previous statement would be beneficial.

Regarding the reviewer's question, our main point here is that higher diterpenoids are emitted in wildfires compared to lab combustion experiments. Our assumption is that heat-induced evaporation and the high abundance of diterpenoids in conifer stems and needles are possible causes of this. Due to the uncontrolled nature of the fires, our samples have different amounts of conifer stems burned and different amounts of unburned fuel heated. But as shown in Figure 3, the emissions of diterpenoids can consistently be predicted by the MCE, despite these differences.

L282: "…are higher than those" how much higher? Quantify w/ percentage or fraction/factor.

We thank the reviewer for this suggestion. It is a bit hard to use a single fraction to describe the differences. We think it is better to put the mean EF values here, which allows for better comparison by the readers. The sentence now reads as follows:

*"The monoterpene EFs, particularly in the Nethker Fire (1.9 g/kg), are higher than those measured in smoke samples from coniferous species (0.3 g/kg on average) burned during the Fire Lab 2016 study (Hatch et al., 2019), with the exception of the canopy samples from subalpine fir, Douglas fir, ponderosa pine and Engelmann spruce, which have summed EFs of monoterpenes above 1 g/kg, higher than the summed monoterpene EF from the Arizona Fires (0.7 g/kg) (Figure S5)."*

L296-299: quantify comparative statements. (e.g values were within X% of one another, these ratios are Y of Z from _____ study).

We thank the reviewer for this suggestion. However, in this case because there is a big difference between the number of samples in the lab (13-15 samples) and the field samples (2-3 samples), we think it would not be very appropriate to compare them using their averages. Also, we think it is not very appropriate to calculate the percentage difference between the log-transformed ratios. Instead, the use of the swarmplot (Figure 6) puts all samples from the field and the lab into perspective and gives the reader a better overview of the sampled fires.

*"As shown in Figure 6, The log-transformed benzonitrile/furfural emission ratio from the wildfires in this study is very close to the ratios of the Blodgett burns. These ratios fall between burning woody fuels and composite fuels (included all individual fuel components) in the Fire Lab 2016 study, indicating the dominance of wood tissue combustion in the wildfires in this study (Figure 6)."*

L304: "significantly higher" again quantify/qualify

This sentence is a citation. The statement is taken from another paper. We do not have the exact numbers. But based on the figure in that paper, we can modify the sentence to:

*"It was reported that 3-carene emission from burning wood is significantly (at least 4 times) higher than burning needles of black spruce and ponderosa pine (Hatch et al., 2015)."*

L314-315: awkward wording, consider rewrite

We would like to simplify the sentence to:

*"This difference suggests that Douglas fir or subalpine fir were burned in the Nethker Fire."*

L316: 'still much smaller' – how much?

Since we have mentioned the fractions of bornyl acetates in our samples, we would like to directly mention the fraction of bornyl acetate in Douglas fir and subalpine fir, reported in Hatch et al. (2019). Now the sentence becomes:

*"However, the abundances of bornyl acetate in the emissions from these wildfires are still much smaller than the source signature of burning Douglas-fir or subalpine fir (in which ~30% of monoterpene emitted is bornyl acetate), which were important plants present in the Nethker Fire and the Castle Fire's perimeters."*

L324: Do you have a sense/proxy for how aged the samples in this study were by comparison? What fraction of what was sampled do you suspect had already formed SOA? Maybe hard to determine

The exact ages of the samples are hard to determine because we only have the daily fire perimeters, but not the exact fire points where the emissions took place. We would like to mention Figure 1 here to allow users to know the distances between the sampling points and the fire perimeters instead. The sentence is revised into:

*"Since our samples were collected close to fires (most samples were taken within 10 km of the fires, as shown in Figure 1), the concentration and contribution of nitrogen-containing compounds to total quantified OA were lower than that observed in BBOA measured 50-60 km downwind from northern CA wildfires (Liang et al., 2021)."*

L328: how much higher? Define statistically significant

We use $p < 0.05$ as the indicator of statistically significant difference. The sentence is revised into:

*"The average nighttime concentration of nitroaromatic compounds were higher by 66% (although not statistically significantly higher, p > 0.05) than in the daytime samples, possibly due to the higher yield of NO$_3$ oxidation (Finewax et al., 2018)..."*

L356-357: rewrite, can't follow

We would like to simplify the sentence into:

*"If the initial emissions for each sample are the same, for each compound, if r > 0, its mass fraction in total quantified increased in aging. If r < 0, its fraction decreased in aging."*

L362: remove "though"

The sentence is change to:

*"4-Nitrocatechol, 3-methyl-5-nitrocatechol and 4-methyl-5-nitrocatechol, with C\*'s near $10^3$ μg m$^{-3}$, have r's of 0.85, 0.77 and 0.79."*

L386: Isn't EC separate from OC? So should say "… phase organic compound EFs, but not for EC"

It is changed as suggested.

L386: "Dominated" – list fractional contribution

The sentence is revised into:

*"Diterpenoids (including resin acids) were the most abundant particle-phase organic compounds detected in the wildfire smoke samples (accounting for 35% of total quantified OA on average), likely due to heat-driven evaporation."*

L388: how much higher?

The emission factor/ratios of monoterpenes are highly variable, as shown in Figures 6 and 7. We think it is not very good to put a number in the conclusion. We revised the text at line 282 (as written in the response above).

L389: How much of a decrease?

As shown in Figures 8 and 9, the decrease in the semivolatile fraction of BBOA is very variable, and it depends on the extent of dilution and many other factors. It may not be a good idea to put a number here to avoid our conclusion being hastily used by the readers without reading the Results and Discussion section.

L391-392: Broad statement, specify how it can contribute to health impacts. E.g., formation of PM downwind a big point

We meant the speciated measurement can help scientists in public health to find compounds of interest for their toxicological studies. The sentences are revised as follows:

*"Such evaporation led to a decrease of the semi-volatile fraction of particle-phase OC when smoke was transported downwind and diluted. The evaporated compounds may react with atmospheric oxidants to form SOA and $O_3$. The emission profiles reported here can benefit future source*

*apportionment or modeling studies and exposure assessments. They can also help researchers focused on public health to identify compounds for more targeted toxicological studies to better understand the health impacts of wildfire smoke. The compounds identified from our GC × GC analyses can also help researchers with one-dimensional GC-MS in compound identification."*

**Tables & Figures:**

Figure 1: How was a "sample" defined? From the figure it looks like a sample refers to an emission pass near the fire for X amount of minutes? Please clarify in figure/text. Was number of minutes (e.g "n") accounted for in samples? Useful to list date samples were taken if all from the same day w/in figure.

We thank the reviewer for the suggestions. We mentioned in the text that these are **hourly** samples in Sections 2.2 and 2.6. We would like to emphasize it again in the caption. Sampling flow rate and sampling time are both considered when we calculated the concentrations. We would like to add the following sentence to Section 2.6 to make sure the readers understand.

*"The masses of compounds in the samples were converted to mass concentration in the atmosphere using the sampling flow rate and duration data."*

The samples in each map were not taken on the same day. We would like to add the date of sampling information in Table 1.

Table 1:

Include date(s) that fire was sampled

The sampling dates are added in Table 1. The sampling start and end times for each filter used in this work are added as a separate Worksheet in the Data Supplement.

| Fire Name | Final Size (km²)[a] | Start Date | Sampling Dates | Fuels | Order/Class[b] | State |
|---|---|---|---|---|---|---|
| **Nethker Fire** | 9.6 | Aug 04, 2019 | Aug 9-17, 2019 | Lodgepole pine, Engelmann spruce, subalpine fir, big sagebrush | Tree dominated, close tree canopy | Idaho |
| **Castle Fire** | 78.4 | July 12, 2019 | Aug 20-22, 2019 | Mixed conifer (e.g., ponderosa pine, Douglas fir) but also dead and downed trees | Tree dominated, open tree canopy | Arizona |
| **Ikes Fire** | 66.4 | July 25, 2019 | Aug 20-22, 2019 | Mixed conifer, grass, and understory | Tree dominated, open tree canopy | Arizona |
| **204 Cow Fire** | 39.1 | Aug 09, 2019 | Aug 25-26, 2019 | Lodgepole pine, mixed conifer, and dead and downed wood | Tree dominated, close tree canopy | Oregon |

Table 2: Include measurement uncertainty or detection limit if appropriate

The detection limits and uncertainties are added into Table 2 as the reviewer suggested. Now Table 2 becomes

|  | GC×GC - particle phase measurement | GC×GC - gas phase measurement |
|---|---|---|
| Thermal Desorption Unit | Gerstel TDS-3, TDSA2 | TurboMatrix 650, Perkin Elmer |
| Thermal Desorption Temperature | 320 °C | 300 °C |
| Column I (volatility) | Restek Rxi-5Sil-MS (60m, 0.25mm i.d., 0.25 µm film thickness) | Agilent DB-VRX (30 m, 0.25 mm i.d., 0.25 µm film thickness) |
| Column II (polarity) | Restek Rtx-200MS (1m, 0.25mm i.d., 0.25 µm film thickness) | Restek Stabilwax (1.5 m, 0.25 mm i.d., 0.5 µm film thickness) |
| Primary Oven Temperature Program | 40 to 320°C at 3.5°C/min, then hold for 5 min at 320°C. | At 35 °C for 5 min followed by a 2.5 °C/min ramp to 165°C, and 10 °C/min to 210 °C with a final hold of 1 min. |
| Secondary Oven Temperature Program | 15 °C higher than primary oven | 10 °C higher than primary oven |
| Mass Spectrometer | HR ToF-MS | ToF-MS |
| Detection limits | ~1 ng for most compounds (e.g., alkanes, acids, anhydrosugars); ~10 ng for very polar compounds (e.g., 2,4-dinitrophenol and 5-nitrovanillin) | Phenolic compounds and nitrogen-heterocyclic aromatic compounds 10-20 ng, others from 0.1-2 ng |
| Uncertainties | ~±10% for compounds exactly matched with a standard compound. ~±30% for compounds quantified by the nearest standard in the same class. Compounds in the "unknown" class have a systematic uncertainty of 200%. | ~±30% |

**Reviewer 2**

This paper shows the results of analysis of gas- and particle-phase organic compounds emitted from western US wildfires using GCxGC coupled with mass spectrometry. The measurements were performed by the Aerodyne Mobile Laboratory (AML) as a part of the FIREX-AQ project. The authors calculated emission factors (EFs) and emission ratios (ERs) for various gas- and particle-phase compounds. Then, those properties were evaluated from the points of correlations with modified combustion efficiency (MCE) and relationship with saturation concentration and so on. I think that this paper includes the latest new results regarding real-world biomass burning and is valuable for researchers in the fields related to atmospheric and environmental sciences. Therefore, I recommend this paper to be published in Atmospheric Chemistry and Physics. But, I have comments to be addressed before publishing. My comments are listed below.

We thank Reviewer 2 for the very constructive comments and suggestions. We improved our analysis and writing based on these comments. Our point-by-point responses and revisions are as follows.

**Major comments:**

The biggest issue of this paper I think is that details of sample treatment and the relationship between sampling and data points in several figures and tables and are not shown clearly. The authors collected 33 hourly samples, but they didn't describe how many samples were collected for individual fires and the corresponding sampling condition (e.g., when the AML was stationary or in transit). Figures 2 and 3 show 9 data points for the Nethker Fire and 2 for the Arizona Fires. How were these data points obtained (if those data points included both samples obtained when the AML was stationary and in transit, how were EFs calculated individually)? I think such details must be described due to credibility of this manuscript.

We thank the reviewer for raising this issue. We included the sampling route and location for all the samples collected near the Arizona Fires and the 204 Cow Fire, and a selection of samples collected near the Nethker Fire in Figure 1. We would like to cite the publication by Majluf et al. (2022) to show the entire traveling route of the AML during this campaign. We added the dates of sampling information in Table 1. We also added a figure for the remainder of non-background Nethker samples in the Supplement (Figure S12). The sampling start and end times for each filter used in this work are added as a separate Worksheet in the Data Supplement.

[Figure]

**Figure S1**2: Locations of the AML (every minute) when the remaining non-background hourly samples (not included in Figure 1) were collected and perimeter of the Nethker Fire.

Each sample is an hourly sample. It is possible that we encountered multiple plumes in one sample. We took the hourly-integrated concentrations to calculate the time-integrated emission factors for that sample. This information is described in Section 2.6.

**Minor comments:**

Line 232: "markedly higher terpenoids+resin acids EF" Here the authors mention terpenoid and resin acid separately. But according to the description in line 131, resin acid is included in terpenoid group. The statement of this should be consistent throughout the manuscript.

We thank the reviewer for pointing this out. We removed the redundant "+resin acids" in Lines 231 and 232. Now the sentences become:

*"The fits of EF vs. MCE for wildfires (this study) and the Fire Lab combustion experiments (Jen et al., 2019) are shown in Figure S3 for particle-phase OC, total quantified OA mass, aromatics, PAHs, sugars, and terpenoids. Although for a given MCE the EF was lower for particle phase-OC and total quantified OA mass, markedly higher terpenoids EF (mainly diterpenoids and resin acids) were measured from the wildfires"*

The relevant subtitle in Figure 3 is also changed accordingly.

[Figure]

Line 284-285: I didn't follow the difference between "the field adjusted EFs" mentioned here and other EFs used in this paper. More detailed description should be added.

The method for calculating the field adjusted EFs are described in Section 2.6 and the Supplement. "To compare the current study with laboratory combustion studies, we also proposed a method to adjust the emission factor based on the emission factor of CO (Supplement Section 3)."

In Supplement, we wrote:

"While laboratory burns offer some advantages over field burns, conditions may be sufficiently different to warrant adjustments of laboratory-derived emission factors (EFs) to better represent field fires. Specifically, in laboratory experiments modified combustion efficiency (MCE) is typically higher than in wildfires. Previous studies have described and applied methods for adjusting laboratory-derived EFs (e.g., Christian et al. (2003), Yokelson et al. (2008), Selimovic et al. (2018)). Most commonly, laboratory EFs for individual compounds or classes of compounds are plotted as a function of MCE and the data are fit using linear regression; the slope and intercept of the linear fit allows calculation of a field-adjusted EF based on the field-derived MCE. This method requires having enough data points for each compound/class of compounds to obtain a robust linear regression. As shown in Stockwell et al. (2015) and Permar et al. (2021), the slope and intercept are dependent on the compound/compound class and thus global fits are not appropriate. Here, a modified approach was applied in which the EF ratio of the compound of interest to CO was averaged between FIREX laboratory and field measurements:

$$\text{EF}_{adjusted} = \frac{1}{2}\left(\frac{\text{EF}_{i,lab}}{\text{EF}_{CO,lab}} + \frac{\text{EF}_{i,field}}{\text{EF}_{CO,field}}\right) \times \text{EF}_{CO,field} \qquad \text{(S1)}$$

where $\text{EF}_{adjusted}$ is the adjusted emission factor, $\text{EF}_{i}$ is the emission of compound i. Averaging the laboratory and field $\text{EF}_{compound}/\text{EF}_{CO}$ accounts for the combustion-type-specific information from both laboratory and field studies, while multiplying by $\text{EF}_{CO,field}$ removes bias from the higher overall MCE (higher flaming/smoldering ratio) in the laboratory studies than in the field. In this work, the field EFs were calculated by averaging across the Arizona and Nethker samples."

Line 327-330: "The average nighttime concentration of nitroaromatic compounds … in FIREX-AQ (Decker et al., 2021)" Is this a result of this work? If so, the corresponding results should be added.

The Decker citation is not a result of this work. It is used for comparison only. We would like to elaborate this point. Now the sentence reads:

*"This is in agreement with the airborne plume study in FIREX-AQ (Decker et al., 2021), which also found the strong oxidation of phenolic compounds by NO$_3$. In our study, the daytime oxidation of phenolic compounds by NO$_3$ is probably less important than in those airborne plumes, possibly due to NO emission on the ground that consumed NO$_3$."*

Line 334-335: "We observed possible SOA marker compounds such as butanedioic acid and octanedioic acid in the samples." How much were those compounds emitted? It would be good if more discussion on the relationship between SOA formation and these compounds can be added.

We want to clarify that our main point here is these compounds have low concentrations in the samples. We would like to revise the discussion as follows:

*"Possible BB SOA marker compounds, dicarboxylic acids such as butanedioic acid (succinic acid) and octanedioic acid (suberic acid) were detected, but their contribution to OC is an order of magnitude lower than in the Napa Fire samples. A later-generation day-time oxidation product of BBOA that is typically observed in aged smoke, malic acid, was not detected in any of the samples (including the background samples)."*

Line 342: "OH" and "NO3" should be "OH concentration" and "NO3 concentration", respectively.

The word concentration is added as suggested. Now the sentence becomes:

*"...OH concentration = 1.5 × 10$^6$ molecule cm$^{-3}$, NO$_3$ concentration = 2.5 × 10$^8$ molecule cm$^{-3}$"*

Figures 2 and 3:

- Why don't the results from the Cow Fire include?

The Cow Fire samples did not meet our criteria for selecting samples for emission factor calculations ($R^2$ between CO and $CO_2 > 0.5$ and CO > 500 ppb). We would like to add one sentence to the figure captions for Figures 2 and 3 to make sure the readers understand what is going on: "*Data points from the 204 Cow Fire did not meet the criteria to be used for EF calculations.*"

- These figures are very similar to Figure S3. Also, according to the story from line 230, it would be good to combine Figure S3 with Figures 2 and 3.

  We thank the reviewer for this suggestion. However, Figure 3 is already very large. It is not easy to squeeze Figure S3 into the same space. It is possible to put the fit from the Jen et al. (2016) study to Figures 2 and 3. However, not all the chemical classes are directly comparable. Adding the fits from Jen et al. to some of the sub-plots may cause some inconsistency. We therefore think it is better to separate these figures.

Figure 6:

- What does color code mean?

- Which markers correspond to the Fire Lab 2016 laboratory or this study? Markers should be shown with different symbols.

We agree with the reviewer that these color codes are not very helpful because the fuel/fire information is already given by the labels of the x-axis. The colors are redundant. We would like to reduce to 3 colors of markers. One color for the Fire Lab data points, one color for the data points from the Blodgett Fires, and one for the FIREX-AQ field data. The updated figure is attached below. The caption is also elaborated.

[Figure]

*Figure 6. Swarm plots of the log-transformed ratio of biomass-burning tracers benzonitrile and furfural for the Fire Lab 2016 laboratory study (shrub, canopy, composite, duff, litter, and wood fuels), the Blodgett study (BFRS) (Hatch et al. 2019) and the wildfires in this study.*

**Technical corrections:**

Line 164: "Organic Carbon (EC)" should be "Organic Carbon (OC)".

We thank the reviewer for catching this error. It is fixed as suggested.

Line 225: "…(Coggon et al., 2016). for the EFs…" Period should not be needed.

We thank the reviewer for catching this error. The typo is corrected.

Line 238: "Simoneit et al., 1993).." One of period is not needed.

We thank the reviewer for catching this error. We deleted it as suggested.

Line 372: "f_44" should be "f_CO2+".

We thank the reviewer for spotting this error. It is changed as suggested.

**References:**

Christian, T. J., Kleiss, B., Yokelson, R. J., Holzinger, R., Crutzen, P. J., Hao, W. M., Saharjo, B. H., and Ward, D. E.: Comprehensive laboratory measurements of biomass-burning emissions: 1. Emissions from Indonesian, African, and other fuels, J. Geophys. Res. Atmos., 108, 4719, https://doi.org/10.1029/2003jd003704, 2003.

Decker, Z. C. J., Robinson, M. A., Barsanti, K. C., Bourgeois, I., Coggon, M. M., DiGangi, J. P., Diskin, G. S., Flocke, F. M., Franchin, A., Fredrickson, C. D., Gkatzelis, G. I., Hall, S. R., Halliday, H., Holmes, C. D., Huey, L. G., Lee, Y. R., Lindaas, J., Middlebrook, A. M., Montzka, D. D., Moore, R., Neuman, J. A., Nowak, J. B., Palm, B. B., Peischl, J., Piel, F., Rickly, P. S., Rollins, A. W., Ryerson, T. B., Schwantes, R. H., Sekimoto, K., Thornhill, L., Thornton, J. A., Tyndall, G. S., Ullmann, K., Van Rooy, P., Veres, P. R., Warneke, C., Washenfelder, R. A., Weinheimer, A. J., Wiggins, E., Winstead, E., Wisthaler, A., Womack, C., and Brown, S. S.: Nighttime and daytime dark oxidation chemistry in wildfire plumes: an observation and model analysis of FIREX-AQ aircraft data, Atmos. Chem. Phys., 21, 16293–16317, https://doi.org/10.5194/acp-21-16293-2021, 2021.

Hatch, L. E., Luo, W., Pankow, J. F., Yokelson, R. J., Stockwell, C. E., and Barsanti, K. C.: Identification and quantification of gaseous organic compounds emitted from biomass burning using two-dimensional gas chromatography-time-of-flight mass spectrometry, Atmos. Chem. Phys., 15, 1865–1899, https://doi.org/10.5194/acp-15-1865-2015, 2015.

Hatch, L. E., Jen, C. N., Kreisberg, N. M., Selimovic, V., Yokelson, R. J., Stamatis, C., York, R. A., Foster, D., Stephens, S. L., Goldstein, A. H., and Barsanti, K. C.: Highly Speciated Measurements of Terpenoids Emitted from Laboratory and Mixed-Conifer Forest Prescribed

Fires, Environ. Sci. Technol., 53, 9418–9428, https://doi.org/10.1021/acs.est.9b02612, 2019.

Liang, Y., Jen, C. N., Weber, R. J., Misztal, P. K., and Goldstein, A. H.: Chemical composition of PM2.5 in October 2017 Northern California wildfire plumes, Atmos. Chem. Phys., 21, 5719–5737, https://doi.org/10.5194/acp-21-5719-2021, 2021.

Majluf, F. Y., Krechmer, J. E., Daube, C., Knighton, W. B., Dyroff, C., Lambe, A. T., Fortner, E. C., Yacovitch, T. I., Roscioli, J. R., Herndon, S. C., Worsnop, D. R., and Canagaratna, M. R.: Mobile Near-Field Measurements of Biomass Burning Volatile Organic Compounds: Emission Ratios and Factor Analysis, Environ. Sci. Technol. Lett., https://doi.org/10.1021/acs.estlett.2c00194, 2022.

Permar, W., Wang, Q., Selimovic, V., Wielgasz, C., Yokelson, R. J., Hornbrook, R. S., Hills, A. J., Apel, E. C., Ku, I. T., Zhou, Y., Sive, B. C., Sullivan, A. P., Collett, J. L., Campos, T. L., Palm, B. B., Peng, Q., Thornton, J. A., Garofalo, L. A., Farmer, D. K., Kreidenweis, S. M., Levin, E. J. T., DeMott, P. J., Flocke, F., Fischer, E. V., and Hu, L.: Emissions of Trace Organic Gases From Western U.S. Wildfires Based on WE-CAN Aircraft Measurements, J. Geophys. Res. Atmos., 126, https://doi.org/10.1029/2020JD033838, 2021.

Selimovic, V., Yokelson, R. J., Warneke, C., Roberts, J. M., De Gouw, J., Reardon, J., and Griffith, D. W. T.: Aerosol optical properties and trace gas emissions by PAX and OP-FTIR for laboratory-simulated western US wildfires during FIREX, Atmos. Chem. Phys., 18, 2929–2948, https://doi.org/10.5194/acp-18-2929-2018, 2018.

Stockwell, C. E., Veres, P. R., Williams, J., and Yokelson, R. J.: Characterization of biomass burning emissions from cooking fires, peat, crop residue, and other fuels with high-resolution proton-transfer-reaction time-of-flight mass spectrometry, Atmos. Chem. Phys., 15, 845–865,

https://doi.org/10.5194/acp-15-845-2015, 2015.

Sumlin, B., Fortner, E., Lambe, A., Shetty, N. J., Daube, C., Liu, P., Majluf, F., Herndon, S., and Chakrabarty, R. K.: Diel cycle impacts on the chemical and light absorption properties of organic carbon aerosol from wildfires in the western United States, Atmos. Chem. Phys., 21, 11843–11856, https://doi.org/10.5194/acp-21-11843-2021, 2021.

Yokelson, R. J., Christian, T. J., Karl, T. G., and Guenther, A.: The tropical forest and fire emissions experiment: Laboratory fire measurements and synthesis of campaign data, Atmos. Chem. Phys., 8, 4497, https://doi.org/10.5194/acp-8-4497-2008, 2008.